# Production of Pigments under Submerged Culture through Repeated Batch Fermentation of Immobilized *Talaromyces atroroseus* GH2

**Juan Pablo Ruiz-Sánchez** [1], **Lourdes Morales-Oyervides** [1], **Daniele Giuffrida** [2], **Laurent Dufossé** [3] and **Julio César Montañez** [1,*]

1 Facultad de Ciencias Quimicas, Universidad Autonoma de Coahuila, Unidad Saltillo, Saltillo 25280, Coahuila, Mexico

2 Dipartimento di Scienze Biomediche, Odontoiatriche e delle Immagini Morfologiche e Funzionali, University of Messina, 98166 Messina, Italy

3 Chimie et Biotechnologie des Produits Naturels & ESIROI Agroalimentaire, Université de la Réunion, F-97744 Saint-Denis, France

* Correspondence: julio.montanez@uadec.edu.mx

**Abstract:** Pigments of natural origin have become a research trend, and fungi provide a readily available alternative source. Moreover, developing novel processes that increase yields, reduce process time and simplify downstream processing is of increased interest. In this sense, this work proposes an alternative for *Talaromyces atroroseus* GH2 biomass re-utilization to produce pigments through consecutive batches using immobilized mycelium. Different support materials were evaluated for pigment production and immobilization capacity. Then, Taguchi's method was applied to determine the effect of four factors related to fungal immobilization and pigment production (inoculum concentration, support density, working volume and support volume). Afterward, process kinetics for pigment production using immobilized cells of *T. atroroseus* GH2 in consecutive batches were evaluated. All evaluated factors were significant and affected pigment production and microorganism growth differently. At improved conditions, immobilization capacity reached 99.01 ± 0.37% and the pigment production was 30% higher than using free cells. Process kinetics showed that the production could continue for three batches and was limited by excessive microorganism growth. Indeed, more studies are still needed, but the immobilization of *Talaromyces atroroseus* GH2 represents a promising strategy for allowing downstream-processing intensification since immobilized biomass is easily removed from the fermentation media, thus paving the way for the further development of a continuous process.

**Keywords:** azaphilone; pigments; immobilization; statistical design

## 1. Introduction

Day by day, the demand for an alternative to synthetic colorants is increasing, mostly due to concerns about their safety since various synthetic pigments possess a potential risk to humans and the environment [1]. Microorganisms provide a readily available alternative source of natural pigments. As opposed to other natural pigments, microbial ones have huge advantages such as rapid growth, easy processing and not depending on weather conditions or season [1].

Filamentous fungi have been used for large-scale production of molecules of interest with high commercial relevance such as enzymes, antibiotics and feed products. In addition, these fungi are known to produce an extraordinary range of pigments, such as carotenoids, melanins, flavins, phenazines and azaphilones [2]. The pigmented compounds produced by filamentous fungi are of interest in many commercial areas. Therefore, research towards developing optimized bioprocesses that lead us to produce various novel

and valued molecules and secondary products through fermentation with less investment is a hot topic [3].

Certainly, the use of microbial pigments dates from about two centuries ago in ancient China. The first strain characterized for red pigment production in fermented rice was *Monascus* sp., in the late 1980s [4]. Since then, *Monascus*-fermented red rice has been used as a traditional food colorant and medicine. Nonetheless, due to the co-production of citrinin along with pigments by some species of *Monascus*, its use as a food additive is not permitted in other countries. Due to this, researchers have been looking for other strains capable of producing *Monascus*-like pigments.

*Talaromyces* spp. produce pigments that are closely related to those produced by *Monascus* spp. but without the production of citrinin [5]. It has been explained that the gene cluster in charge of the azaphilone pigment production is widely distributed among various filamentous fungi genera, viz. *Monascus*, *Talaromyces* and *Penicillium* [5].

Indeed, the developed bioprocess requires meeting the criteria of product quality, scalability and robustness. In this sense, developing a cost-effective process with high pigment production yield is crucial. Different bioprocessing strategies have been elucidated in the literature for reducing bioprocess costs, such as process optimization [6], reducing processing time by implementing a continuous process [7] and using cheap residues as substrates [8].

Regarding pigments produced by *Talaromyces* strains, numerous attempts have been made to optimize the crucial variables affecting the process [9]. Nevertheless, most of the studies have been carried out in batch cultivation. Pigments produced by *Talaromyces atroroseus* are intracellular and extracellular but are mainly excreted out of the cell and into the producing media [9]. Therefore, cell immobilization for recycling biomass through repeated batches represents an alternative for developing a continuous process.

Immobilization techniques have been applied over microorganisms, isolated microbial enzymes and cells to synthesize various high-value compounds [10]. Cell immobilization strategies have been demonstrated to be an impactful method regarding the economics of a whole bioprocess [11].

The utilization of immobilized biomass has been proven to increase the biomass resistance to thermal, chemical and shear forces; facilitate biomass recycling for a continuous process; and reduce the time for the downstream processes such as solid–liquid separation, recovery of the product of interest and filtration [12]. Accordingly, since the pigments produced by *Talaromyces* spp. are released and diffused on the liquid media, having the biomass immobilized onto a support permits easily separating the support with the biomass, leaving the diffused pigments on the fermentation media. All these advantages represent a reduction in the overall costs of the process.

Moreover, the ability of fungi to attach to various surfaces gives them a higher capacity to colonize and overcome harsh environmental conditions [13]. Fungal biomass immobilization is governed by biological and physical parameters such as proteins located in the fungal cell membrane and hydrophobic interactions between fungal cells and surfaces where immobilization occurs. These interactions permit the fungi to attach to biotic (plants, other species, etc.) or abiotic (inert or synthetic supports) surfaces. This natural behavior has been investigated to intentionally immobilize cells in specific supports for easy control and reuse [14]. In addition, it is known that immobilized filamentous fungi have an increased capacity for secondary metabolite biosynthesis [15].

The nature of the selected support for the immobilization plays a crucial role in the process. In this sense, one of the advantages of the utilization of synthetic/inert supports over natural supports is that the microorganism does not consume it. For instance, natural supports may deliver and alter the composition of the medium as the action of natural decomposition or biological degradation for the activity of the microorganisms itself, thus adding more variability in the process [16]. Optimization techniques must be applied to maximize processing time, cost-effectiveness and metabolite-of-interest production. In this work, the Taguchi methodology was selected to determine the influence of factors that can

affect the microorganism's immobilization during the pigment production process. The Taguchi method has been widely applied at an industrial level due to its ability to improve the process by performing a relatively small number of experimental runs. This method was previously applied to select the best inoculum that enhances pigment production by *Talaromyces atroroseus* GH2 [17].

First, various inner supports were evaluated to depict its fungal biomass immobilization capacity. The immobilization conditions were then improved, and the process kinetics were studied in either a single batch manner or a repeated batch by recycling the immobilized biomass.

## 2. Materials and Methods

### 2.1. Microorganism

*Talaromyces atroroseus* GH2 was used for immobilization before pigment production. The purified strain had been previously isolated and characterized as *P. purpurogenum* GH2. Nevertheless, *P. purpurogenum* has been transferred to *Talaromyces* spp. [18] and recently identified as *Talaromyces atroroseus* GH2. Fungal spores were stored at $-20\,^{\circ}$C in a cryo-preservative solution of skimmed milk and glycerol.

### 2.2. Media

The Potato Dextrose Broth (PDB, ATCC medium: 336) was prepared by heating 50 g/L of dehydrated potato flakes at 85 $^{\circ}$C for 30 min; the mixture was then filtered through cheesecloth. Finally, 20.0 g/L of glucose was added before sterilization. Potato Dextrose Agar (PDA) was prepared equally to the PDB medium but with the addition of agar (15 g/L). The Czapek–Dox-modified (CDM) medium reported by Mendez-Zavala [19] for pigment production consisted of (g/L) D-xylose (15.0), NaNO$_3$ (3.0), MgSO$_4\cdot$7H$_2$O (0.5), FeSO$_4\cdot$7H$_2$O (0.1), K$_2$HPO$_4$ (1.0), KCl (1.0) and ethanol (20.0). CDM medium initial pH was adjusted to 5 with a HCl solution before sterilization using 0.22 μm sterile membranes (Millipore, Billerica, MA, USA).

### 2.3. Inoculum

For spore preparation, the *Talaromyces atroroseus* GH2 strain was grown in 250 mL Erlenmeyer flasks with 50 mL of PDA medium and incubated at 30 $^{\circ}$C for 120 h. After the incubation time, a suspension of spores was obtained by washing the cultures with a sterile aqueous solution of Tween 20 (0.01% *v*/*v*) with the help of a magnetic stirrer (Heidolph Instruments GmbH & CO. KG, Schwabach, Germany).

A mycelial suspension was used as inoculum prepared in 250 mL Erlenmeyer flasks containing 50 mL of PDB medium. Flasks were sterilized and inoculated with the previously prepared spore suspension ($1 \times 10^5$ spores/mL) of *Talaromyces atroroseus* GH2. The flasks were incubated at 30 $^{\circ}$C for 72 h in an orbital shaker (Inova 94, New Brunswick Scientific, Edison, NJ, USA) at 200 rpm.

### 2.4. Support Selection for Immobilization

Two nylon sponges (Delicate Duty, A; Kitchen Heavy Duty Grill, B) and one stainless-steel sponge (C) (all Scotch-Brite, 3M Spain, SA) were tested for their capacity for fungal biomass immobilization and pigment production. Before use, cubes of nylon sponges and irregular cuttings of the stainless-steel sponge were pre-treated, as reported by Castro et al., with some modifications [20].

Briefly, the nylon sponge cubes and stainless-steel sponge cuttings were boiled for 10 min and washed thoroughly three times with distilled water. Afterward, the supports were oven-dried overnight at 60 $^{\circ}$C. Before biomass immobilization, the carriers were autoclaved at 121 $^{\circ}$C for 15 min.

Pigment production with immobilized biomass was carried out in Erlenmeyer flasks (125 mL) containing each evaluated support. The support volume was 1 cm$^3$ for A and B and irregular cuttings for the stainless-steel sponge. The same numbers (11) of cubes

and stainless-steel pieces were used; thus, the supports' density (support weight/working volume) differed for A (0.03 g/mL), B (0.07 g/mL) and C (0.13 g/mL). Erlenmeyer flasks containing the supports were sterilized (121 °C, 15 psi, 15 min), and sterile CDM medium was added (25 mL). The inoculated flasks (10%) were incubated at 30 °C on an orbital shaker (Inova 94, New Brunswick Scientific, Edison, NJ, USA) at 200 rpm for 168 h.

At the end of the first batch, the medium containing the extracellular pigments was recovered and the supports with immobilized biomass were aseptically transferred to a new flask with fresh CDM medium (Section 2.7). Culture conditions were kept as described in the paragraph above. The process was evaluated in three consecutive batches. Pigment production (P, $OD_{500nm}$) and loosened biomass ($X_L$, g/L) were assessed for the 1st, 2nd and 3rd batches. Total biomass (X, g/L) and immobilization capacity ($I_M$, %) were evaluated only at the end of the 3rd batch. All experiments were carried out in triplicate.

### 2.5. Improvement of Immobilization Conditions

After support selection, Taguchi's method was applied to determine the individual and combined effects of four factors related to the fungal immobilization capacity to increase pigment production. The evaluated factors were inoculum concentration (% *v/v*), support density (g/mL), working volume (mL) and support volume ($cm^3$). Evaluated levels were designated after support selection and thus will be described in the relevant results section. Evaluated responses were pigment production, biomass growth, immobilization capacity and pigment per biomass yield. Once improved conditions were selected, a validation trial was carried out under selected settings in a single batch to infer possible interactive effects between factors.

### 2.6. Kinetics Evaluation under Improved Immobilization Conditions

Process kinetics of pigment production, biomass growth and substrate consumption were evaluated under improved conditions. Process kinetics of free biomass were also monitored for a single batch. Culture conditions for free and immobilized biomass were the same as previously described. Samples were taken every 24 h ("destructive samples") during a single batch (168 h) for analysis.

After evaluating the kinetics in a single batch, process kinetics of pigment production, biomass growth and substrate consumption were evaluated in three consecutive batches with selected support at improved immobilized conditions. Samples were taken every 24 h ("destructive samples").

### 2.7. Analytical Methods

The medium containing the extracellular pigments was separated from the immobilized biomass with the help of a syringe. Thereafter, pigment recovery was performed according to the methodology reported by [21]. The pigment extract was centrifuged at 8000 rpm and 4 °C for 20 min (Sigma-18KS, Sigma Laborzentrifugen GmbH, Osterode am Harz, Germany) and then filtered through 0.45 μm cellulose membranes (Millipore, Billerica, MA, USA) prior to analysis. In this study, only extracellular pigments were considered.

The amount of red pigments produced by *Talaromyces atroroseus* GH2 has been reported by different authors as optical density units measured at 490–500 nm (a wavelength that represents the maximum adsorption for red colorants) [18], and pigment amount was quantified indirectly by simply measuring the optical density at 500 nm using a spectrophotometer (Unico UV 2150, Unico, Fairfield, NJ, USA).

The immobilized biomass dried weight was determined based on the difference between the weight of the support after fermentation and the weight of the support itself before fermentation [20]. The loosened biomass ($X_L$, g/L) was determined using the gravimetric method. Immobilization capacity ($I_M$, %) was calculated based on the percentage of immobilized biomass ($X_I$, g/L) and total biomass (X, g/L) [20]:

$$I_M = \left( \frac{X_I}{X_I + X_L} \right) \times 100 \qquad (1)$$

Substrate consumption was analyzed by quantifying the total sugar content using the method reported by Dubois [22]. Pigment yield ($OD_{500nm}/g/L$) was defined as the extracellular pigment per biomass obtained at the end of the batch.

### 2.8. Data Analysis

Target responses for screening support and continuous process assessment (pigment production, biomass growth and immobilization capacity) were analyzed with an ANOVA to test statistical differences ($p < 0.05$), followed by post hoc analysis with Tukey's test at 5% probability to define homogeneous groups.

For the improvement of immobilization conditions, single responses (pigment production, biomass growth, pigment per biomass yield and immobilization capacity) were first analyzed with a two-way analysis of variance (ANOVA). If all interactions are negligible, then the level for each factor capable of maximizing the target response is the one yielding the highest average response [23].

The confidence interval of the predictions for a 90% confidence level was calculated as previously reported [23].

In improved immobilized conditions and free biomass (control), fermentation kinetics were studied and described by unstructured models for a single batch.

Biomass growth rate (dX/dt, g/L.h) was described by the logistic model [24]:

$$\frac{dX}{dt} = \mu_o \left( 1 - \frac{X}{X_{max}} \right) X \tag{2}$$

where $\mu_o$ is the initial growth rate, $X_{max}$ represents the maximum obtained biomass (g/L) and X is the instantaneous biomass concentration (g/L).

The Luedeking–Piret model was used to describe the rate of pigment production (dP/dt, OD/h) [25]:

$$\frac{dP}{dt} = \alpha \frac{dX}{dt} + \beta X \tag{3}$$

The model includes two parameters with significant biological meaning, $\alpha$ for production related to growth and $\beta$ for production related to maintenance. The substrate consumption was calculated as a function of the utilization of substrate for biomass growth, pigment production and cell maintenance [26]:

$$\frac{dS}{dt} = -\gamma \frac{dX}{dt} - \eta X \tag{4}$$

where

$$\gamma = \frac{1}{Y_{X/S}} + \frac{\alpha}{Y_{P/S}} \tag{5}$$

and

$$\eta = \frac{\beta}{Y_{P/S}} + k_e \tag{6}$$

where $Y_{X/S}$ is the yield of biomass per substrate ($g_{Biomass}/g_{Susbstrate}$), $Y_{P/S}$ is the yield of pigments per substrate ($OD/g_{Susbstrate}$) and $k_e$ is the maintenance coefficient.

## 3. Results and Discussion

### 3.1. Support Selection for Immobilization

At first insight, the immobilization of *Talaromyces atroroseus* did not imply a significant increment if compared with the production using free mycelium ($p$, $10.0 \pm 0.5$ $OD_{500nm}$). Nevertheless, the microorganism could grow and produce pigments among all evaluated supports. Table 1 shows the results of pigment production and loosened biomass in three consecutive batches and the final total biomass production and immobilization capacity in each support. Homogeneous groups were assigned to indicate statistical differences between supports A, B and C for each batch and for each response. Unfortunately, we

only observed higher pigment production ($p < 0.05$) using support C during the first batch, and this was associated with the high experimental error. In any case, this preliminary evaluation aimed to select a suitable support for mycelial cells of *Talaromyces atroroseus* GH2 with high immobilization capacity. Yet, again $I_M$ was not statistically different between the supports ($p > 0.05$). Thus, other considerations were made for support selection. Support C was first ruled out due to the lower average immobilization capacity at the end of the third batch, which means leakage of biomass batch to batch. Even if unstable, both A and B supports kept the pigment production activity during the three batches and had a similar retention percentage, although support B exhibited the most unstable retention ability of these three evaluated supports.

**Table 1.** Pigment production and total and loosened biomass in three consecutive batches using three different supports.

| Batch | Supports | | | | | | | | | | | |
|---|---|---|---|---|---|---|---|---|---|---|---|---|
| | A | | | | B | | | | C | | | |
| | $P$, $OD_{500nm}$ | $X_L$, g/L | $X$, g/L | $I_M$, % | $P$, $OD_{500nm}$ | $X_L$, g/L | $X$, g/L | $I_M$, % | $P$, $OD_{500nm}$ | $X_L$, g/L | $X$, g/L | $I_M$, % |
| 1 | [b] 3.00 ± 1.12 | [a] 4.97 ± 1.97 | - | - | [b] 2.88 ± 3.43 | [b] 0.48 ± 0.25 | - | - | [a] 8.63 ± 0.69 | [b] 1.35 ± 0.24 | - | - |
| 2 | [a] 9.54 ± 4.99 | [a] 3.55 ± 1.18 | - | - | [a] 2.41 ± 1.51 | [a] 1.12 ± 0.66 | - | - | [a] 15.86 ± 9.98 | [a] 2.91 ± 1.07 | - | - |
| 3 | [a] 2.68 ± 0.71 | [a] 1.85 ± 0.48 | [a] 13.01 ± 2.02 | [a] 85.90 ± 1.37 | [a] 2.73 ± 2.40 | [a] 2.00 ± 1.42 | [a] 13.68 ± 1.88 | [a] 85.35 ± 10.03 | [a] 4.56 ± 2.36 | [a] 2.96 ± 1.01 | [a] 14.76 ± 1.50 | [a] 79.73 ± 7.17 |

A, B and C stand for the evaluated supports. Pigment production (P), total biomass (X), loosened biomass ($X_L$), immobilization capacity ($I_M$). Superscripts [a, b] indicate homogeneous groups (Tukey's test at 5% probability).

　　Regarding the morphology of the immobilized biomass, two of these supports exhibited a similar morphology regarding biomass attachment and growth; this was the case of supports A and B, which exhibited an internal matrix fully occupied by the mycelial cells. For support C, even though the biomass could penetrate the support, it kept growing on the outer surface and provoked a biomass loss during the process.

　　Moreover, supports B and C presented higher experimental variability in terms of $I_M$, which was undesired. Hence, only Support A was selected for further improvement of immobilization conditions based on the observed pigment production and immobilization capacity as seen in Figure 1. Choosing the right immobilization support directly impacts the microorganism metabolism because it can influence a good immobilization performance and a correct mass transfer into the support and through the biomass [27]. The nature of the selected support for the immobilization plays a key role in the process [28]. Figure 1 shows the selected support before and after the fermentation.

*3.2. Improvement of Immobilization Conditions*

　　In accordance with the results presented in Section 3.1, only support A, Scotch-Brite Delicate Duty Scrubbing Pad, was used in the following experiments.

3.2.1. Taguchi Experimental Design L9 Results

　　After support selection, the Taguchi experimental design was applied due to the high variability observed in the previous stage regarding pigment production and immobilization capacity. It has been reported that the Taguchi method is useful for improving process performance and minimizing variability [17].

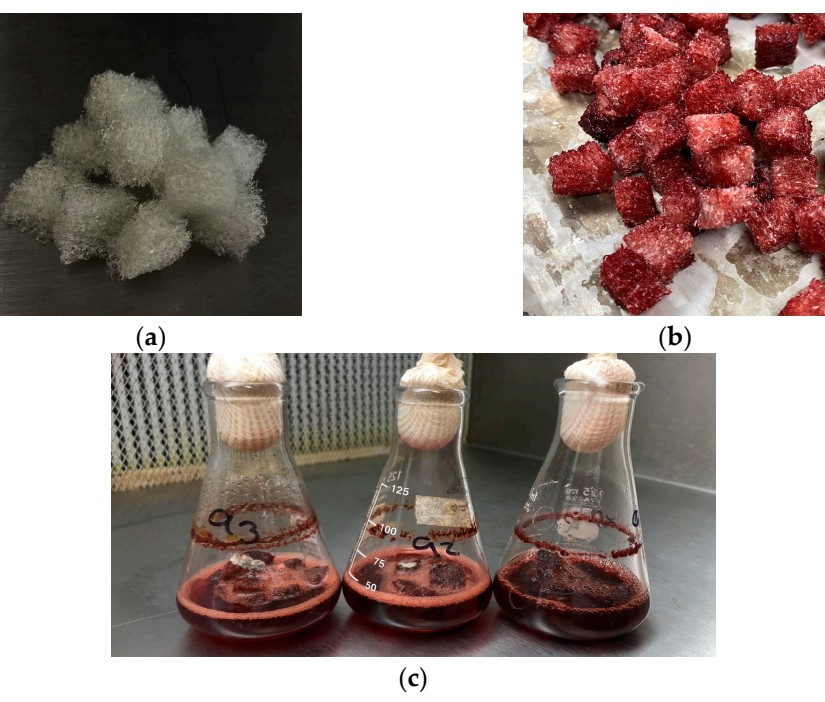

**Figure 1.** Selected support A. (**a**) Support A without biomass prior to fermentation; (**b**) support A with immobilized biomass after the fermentation process; (**c**) flasks with immobilization supports after fermentation.

The effect of four different factors at three levels on pigment production and biomass growth exhibited by *Talaromyces atroroseus* GH2 was evaluated. Table 2 shows the Taguchi L9 orthogonal array and the average results in pigment production, biomass, pigment per biomass yield and immobilization capacity. All the values chosen significantly impacted pigment production, varying from 0.02 to 8.089 $OD_{500nm}$; total biomass growth, from 3.04 to 7.80 g/L; and immobilization capacity, from 72.68 to 100.00%. The global average of pigment production obtained at all evaluated conditions was 3.98 $OD_{500nm}$.

**Table 2.** Experimental design and results of L9 array for immobilization process improvement.

| | **Actual** | | | | **Experimental Results** | | | |
|---|---|---|---|---|---|---|---|---|
| **Run** | **Mycelium, % *v/v*** | **Support Density, g/mL** | **Working Volume, mL** | **Support Volume, cm³** | **P, $OD_{500nm}$** | **$I_M$, %** | **X, g/L** | **Y, OD.L/g** |
| 1 | 5 | 0.01 | 20 | 0.25 | 0.02 ± 0.03 | 93.50 ± 2.82 | 4.34 ± 0.14 | 0.00 ± 0.01 |
| 2 | 5 | 0.024 | 25 | 0.5 | 1.55 ± 0.35 | 92.83 ± 2.39 | 3.04 ± 0.17 | 0.51 ± 0.14 |
| 3 | 5 | 0.03 | 30 | 1 | 7.94 ± 0.03 | 100.00 ± 0.00 | 7.80 ± 0.28 | 1.02 ± 0.03 |
| 4 | 10 | 0.01 | 25 | 1 | 3.00 ± 1.81 | 72.68 ± 0.52 | 5.42 ± 0.31 | 0.56 ± 0.37 |
| 5 | 10 | 0.024 | 30 | 0.25 | 3.50 ± 1.41 | 94.56 ± 0.12 | 6.62 ± 0.14 | 0.53 ± 0.20 |
| 6 | 10 | 0.03 | 20 | 0.5 | 8.89 ± 2.67 | 96.72 ± 2.01 | 4.26 ± 2.40 | 2.69 ± 2.15 |
| 7 | 15 | 0.01 | 30 | 0.5 | 3.53 ± 0.69 | 45.73 ± 4.93 | 7.60 ± 0.17 | 0.47 ± 0.10 |
| 8 | 15 | 0.024 | 20 | 1 | 4.70 ± 0.48 | 81.80 ± 9.82 | 6.44 ± 0.57 | 0.73 ± 0.01 |
| 9 | 15 | 0.03 | 25 | 0.25 | 2.66 ± 1.75 | 83.15 ± 10.22 | 6.98 ± 0.71 | 0.37 ± 0.21 |

Pigment production (P), immobilization capacity ($I_M$), biomass (X), pigment yield (Y).

Run 1 exhibited practically no pigment production and lesser total biomass production than the rest. Runs 2 and 9 had a higher pigment production than run 1, yet lower pigment production than the rest of the runs. Runs 4, 5, 7 and 8 exhibited production of pigments close to the global average, but the immobilization capacity varied greatly among such runs. On the other hand, runs 3 and 6 exhibited augmented pigment production (7.94 ± 0.03 and

$8.89 \pm 2.67$ OD$_{500nm}$, respectively) while maintaining a proper biomass immobilization ($96.72 \pm 2.01$ and $100.00 \pm 0.00\%$).

This pigment production is lower in comparison with that of the same strain in another study [20], although it is important to state that in that study, the highest production ($11.45 \pm 0.17$ OD$_{500nm}$) was reached using free cells.

There is a considerable difference in the biomass immobilization capacity between runs 3 and 6. While trial number 6 exhibited $96.72 \pm 2.01\%$, the immobilization capacity of trial 3 is practically 100%. During trial 3, culture media remained clear with no turbidity produced by a biomass leakage, which indicates a good and lasting immobilization; any leakage could represent an operational problem such as clogs and less activity [29].

While the overall pigment production by *Talaromyces atroroseus* GH2 once immobilized is low, it is worth mentioning that such conditions must be optimized for better results, as shown by de Oliveira et al., who used a similar strain and obtained low pigment production before any optimization [30].

Additionally, the support's possible maximum immobilization capacity was identified as a function of biomass loading per support weight (Figure 2). It was observed that above 0.5 g of biomass per gram of support, the immobilized biomass is significantly reduced. This maximum support capacity to retain the mycelium should be considered for future cell recycling in consecutive batches. Additionally, it has been reported that cell loading per support volume or weight can significantly affect microbial process productivity [31].

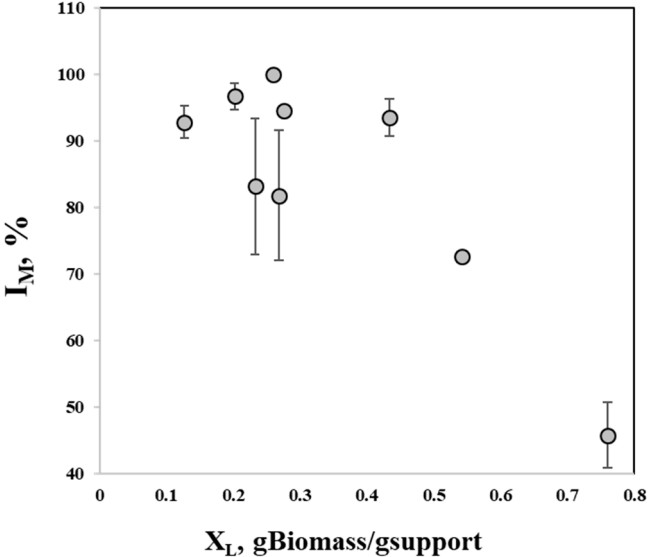

**Figure 2.** Biomass loading ($X_L$, g$_{Biomass}$/g$_{support}$) effect on immobilization capacity. Error bars depict the standard deviation between three replicates.

However, we could not identify a possible correlation between biomass loading and pigment production. In any case, the significance of factors can only be withdrawn from the analysis of the variance of the results.

### 3.2.2. Analysis of the Relative Influence of Factors

Figure 3 shows the ANOVA results depicted as the relative influence (%) of each factor (support volume, working volume, support density and mycelium) over the responses analyzed.

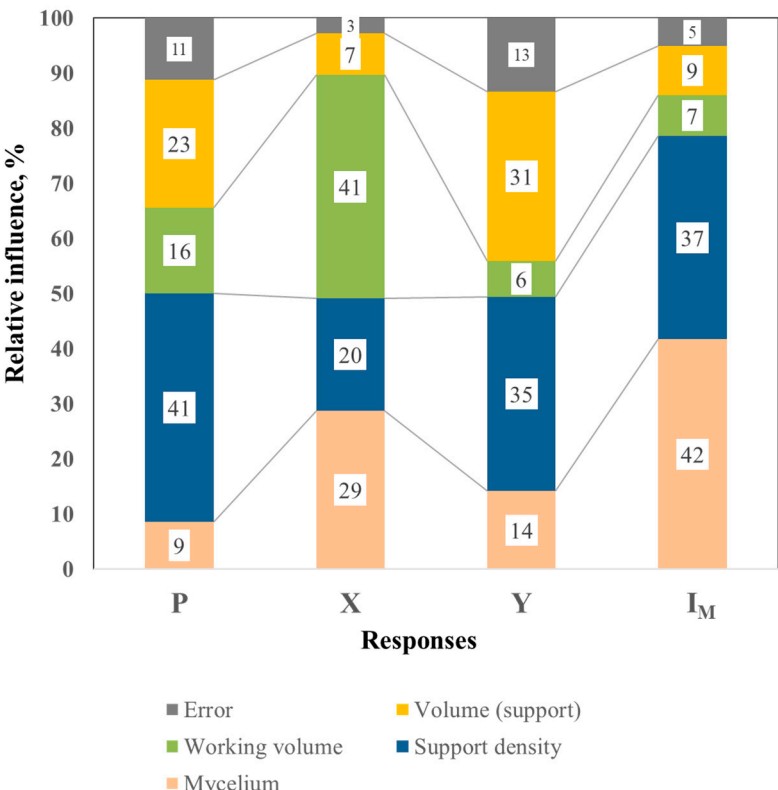

**Figure 3.** Relative influence percentage of the factors over the responses pigment production (P), biomass (X), pigment per biomass (Y) and immobilization capacity ($I_M$).

The support density had the highest relative influence (41%) on pigment production, followed by the utilized support volume (23%). Interestingly, the factors related to support characteristics showed the most significant effect on the pigments produced by immobilized biomass of *T. atroroseus* GH2. In addition, the error explained 11% of the obtained results for this response, which could be attributed to uncontrolled noise variables or interactive effects between factors.

Regarding biomass production, all the factors significantly affected microorganism growth ($p < 0.05$). For this response, the working volume had the greatest influence (41%), followed by the inoculum concentration (29%). In addition, for biomass growth, the error only explained 3% of the results, which implied that this response was less influenced by uncontrolled variables or interactive effects between factors. Concerning the pigment per biomass yield, only the support density and support volume were significant (35% and 31%, respectively).

This confirms that the physical properties of immobilization support are of major importance for both biomass growth and pigment production and that such behavior relies on the correct working volume to positively increase these responses due to the correct working volume increasing the accessibility of nutrients [32]. As for the immobilization capacity, all the evaluated factors were significant, and the inoculum concentration and support density significantly influenced this response (42% and 37%, respectively).

It has been demonstrated how increasing or decreasing cellular concentration in immobilization support directly affects the production of secondary metabolites [10,33]. In addition, the support volume and the working volume explained only a combined effect of 16% on the immobilization capacity. For this response, the error was relatively low (5%), implying that there is no interactive effect between factors on the fungal immobilization capacity of the support.

From the above results, it was evident that all the evaluated factors affected all the responses differently. Mostly, the analysis revealed that the most significant factors affecting

pigment production are not the most important for microorganism growth. Although *Talaromyces* spp. immobilization on inert supports for the production of pigments has not been evaluated yet, our results are consistent with previous studies highlighting that pigment production by *Talaromyces* spp. is not related to growth [32]. It is a risk that the utilized support might emulate the fungal natural environment with proper conditions for growth, resulting in inhibition of pigment production.

In this work, the main target is to select the conditions that will allow a high immobilization of the strain so that the biomass can be recycled for the continuous process mode. Of course, maintaining the microorganism in a secondary metabolism state is crucial.

### 3.2.3. Analysis of Individual Factors

Figure 4 shows the mean plots (averages of all data obtained with each level of each factor), from where the identification of the improved conditions is straightforward: in each case, the best level is the one with the highest mean. It also gives a better understanding of how the responses are affected at the evaluated levels; we focused mainly on the effect of the most significant factors. Pigment production increased linearly with the support density (factor with the highest significance). In addition, a slightly quadratic effect was observed for the support volume. This means that using a higher support weight per working volume and a higher support volume induced the microorganism to produce pigments. Results could be attributed to the requirements of mass transfer into the support and through the biomass [27].

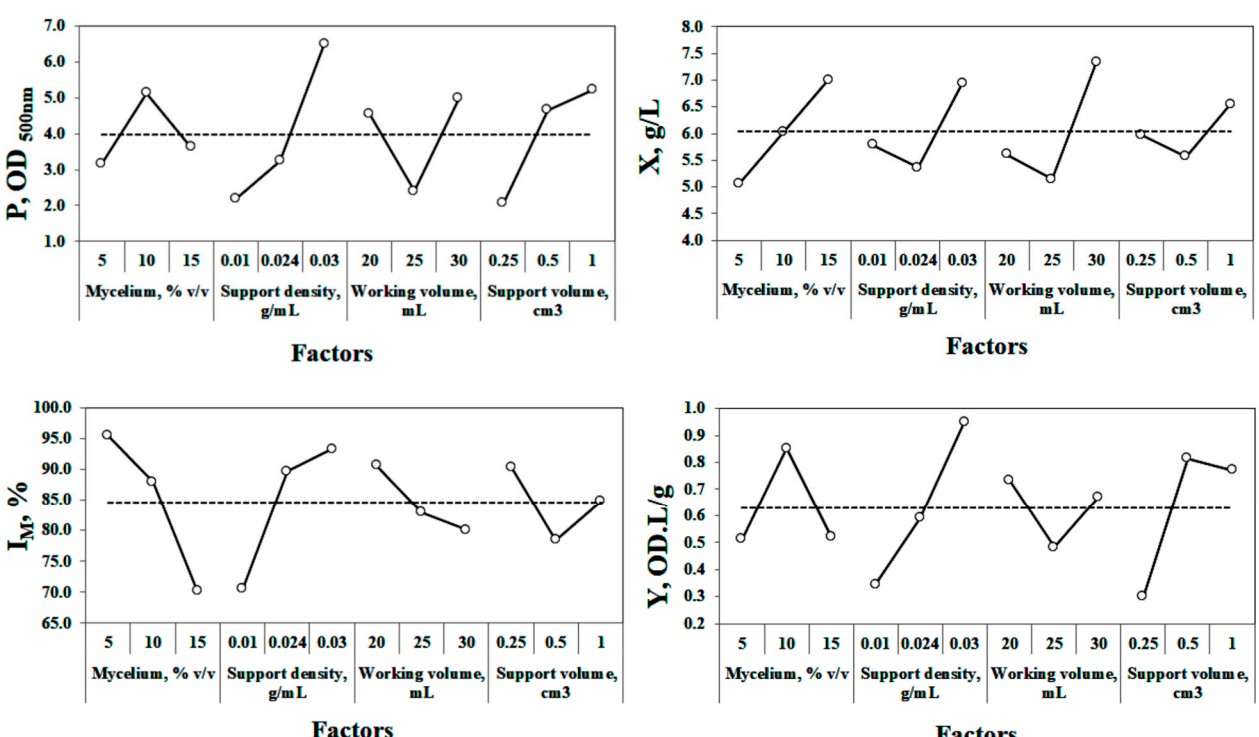

**Figure 4.** The effects of individual factors on each response. Pigments (P), biomass (X), immobilization capacity ($I_M$), pigment yield (Y).

Mass transfer plays a crucial role in the bioconversion rate of immobilized biomass [34]. Authors have described the systems utilizing immobilized biomass in inert support submerged in liquid media as heterogeneous [35]. They describe the existence of two phases, liquid and solid (also named external and internal, respectively). The liquid phase comprises the media where nutrients are dissolved and available, while the solid phase comprises the inner support with the biomass immobilized in and within its matrix.

In heterogeneous systems (solid immobilization supports in the liquid media), the mass transfer rate from the outside to the inside is important. The poor mass transfer may reduce substrate intake and transformation rates. The present and dissolved nutrients in the liquid phase (culture media) must be transferred on the surface and through the supports' immobilized biomass. This flow of nutrients from one phase to another is fundamental and determines the overall reaction rate, thus determining the whole performance of the bioprocess using immobilized biomass. Indeed, analyses such as scanning electron microscopy (SEM) will help describe and explain how immobilization occurs.

In the literature, it has been explained how it is possible to reduce the mass transfer resistance in both solid and liquid phases. For solid-phase mass transfer, the resistance can be reduced by decreasing the support size for biomass immobilization [36]. In contrast, increasing the agitation can reduce liquid-phase mass transfer resistance.

Agitation rates and immobilization support volume have been evaluated to address mass transfer phenomena in immobilized biomass-based reactors [37,38]. It was found that both parameters played a crucial role in the substrate intake and conversion rate, as mentioned above. On the same subject, even though reducing the support size has been proven to reduce the solid-phase mass resistance, thus increasing the conversion rate, other parameters limit the modification of the support size, such as pore size and interaction between the microbial cell and the support matrix [16].

Concerning biomass growth, working volume, the most significant factor, had a positive quadratic effect with an optimum at the highest evaluated working volume. In contrast, the inoculum concentration had a positive linear effect. This confirms that the physical properties of immobilization support are of major importance for biomass growth and that such behavior relies on the correct working volume to increase the accessibility of nutrients positively [34]. As for the linear increase in biomass growth with an increase in inoculum concentration, it could be evident if the nutrients are sufficient, which was the case.

As for Y, the support density presented a clear positive linear effect, while the support volume showed a negative quadratic effect. This response is important to maximize because it would be preferable to have a process with less biomass but maintaining a high pigment production. A process with excess biomass will affect the immobilization, and thus, loosened biomass could affect the hydrodynamics, viscosity of the medium, final productivities and downstream processing. However, Y trends and improved levels mostly correlated with the results obtained for pigment production.

As for the immobilization capacity, it was observed that the immobilization was highly reduced by increasing the inoculum concentration; specifically, shifting the inoculum percentage from 10 to 15% resulted in reduced immobilized biomass. This effect is opposite to biomass growth. Thus, it was assumed that a higher inoculum level resulted in higher biomass growth, which exceeded the support capacity to immobilize the cells, as shown in Figure 3.

Concerning the effect of support density, this factor presented a slightly quadratic effect with an optimum at the highest level. It is interesting to note the difference between the support density and support volume effects. Results showed that more cubes of lower volume were better for *Talaromyces atroroseus* GH2 immobilization than fewer with higher volume. This could be attributed to the ability of the mycelium to penetrate the support structures.

In any case, if all interactive effects are negligible, then the optimum level is the one with the highest average performance. As previously mentioned, our main target is to keep the microorganism in a secondary metabolism state but with a high level of immobilized biomass so that the biomass can be utilized in consecutive batches.

### 3.2.4. Validation Trials

Improved levels for increasing pigment production are an intermediate inoculum concentration (10% $v/v$), the highest support density (0.03 g/mL), and the highest working

volume (30 mL) and support volume (1 cm$^3$). Table 3 shows the expected responses (P, X, Y, I$_M$) at such conditions and the contribution (positive or negative) to the grand average by each factor. The expected response within a confidence interval (minimum and maximum) is also shown. It can be observed that there is only a negative contribution of the factors to the immobilized biomass under the improved conditions for pigment production. The working volume gives this negative contribution; however, it is lower than 5%.

**Table 3.** Contribution of improved settings to each evaluated target, expected response within a confidence level and experimental result at improved conditions.

| Factor | | Responses | | | |
| --- | --- | --- | --- | --- | --- |
| | | P, OD$_{500nm}$ | X, g/L | Y, OD.L/g | I$_M$, % |
| Contribution | Mycelium, % *v/v* | 1.15 | 0.00 | 0.22 | 3.43 |
| | Support density, g/mL | 2.52 | 0.91 | 0.32 | 8.74 |
| | Working volume, mL | 1.01 | 1.31 | 0.04 | −4.45 |
| | Support volume, cm$^3$ | 1.24 | 0.52 | 0.14 | 0.27 |
| Expected response | Minimum | 8.15 | 8.32 | 1.08 | 85.81 |
| | Maximum | 11.65 | 9.23 | 1.64 | 99.28 |
| | Experimental | 13.17 ± 1.52 | 9.68 ± 0.47 | 1.36 ± 0.15 | 99.01 ± 0.37 |

Pigment (P), biomass (X), pigment yield (Y), immobilization capacity (I$_M$).

In any optimization assessment, validating the optimum settings is crucial to validate the methodology. A validation trial was carried out with the previously mentioned improved levels, and the results are also shown in Table 3. There was a satisfactory agreement between the confidence interval of the expected response and the one obtained after the validation trial for X, Y and I$_M$. However, the experimental pigment production after the validation trial was 13.04% higher than the maximum limit expected.

This lack of agreement could be attributed to interactive effects between factors. If we recall the previously mentioned relative influence of factors, 11% of the results are explained by the error.

The importance of the interactive effects between factors relies on the fact that one can be misled to sub-optimum regions, which was not the case. Although there was no perfect agreement for pigment production, the results were higher than expected. Thus, improved conditions for pigment production were selected for the subsequent studies. Both high pigment production and high biomass immobilization can be attained under these conditions.

*3.3. Process Kinetics Evaluation*

3.3.1. Process Kinetics and Model Fitting Using Free and Immobilized Biomass

Figure 5 shows the kinetic behavior of free and immobilized cells under improved conditions of *Talaromyces atroroseus* GH2 in a single batch fashion regarding pigment production, biomass and substrate consumption. Table 4 displays the model parameters of equations presented in Section 2.8 (Equations (4)–(6)). It can be observed that the models presented a good correlation coefficient ($R^2 > 0.9$) in all cases except for biomass growth with free cells. Those results are attributed to the logistic model's inability to describe the cell death phase.

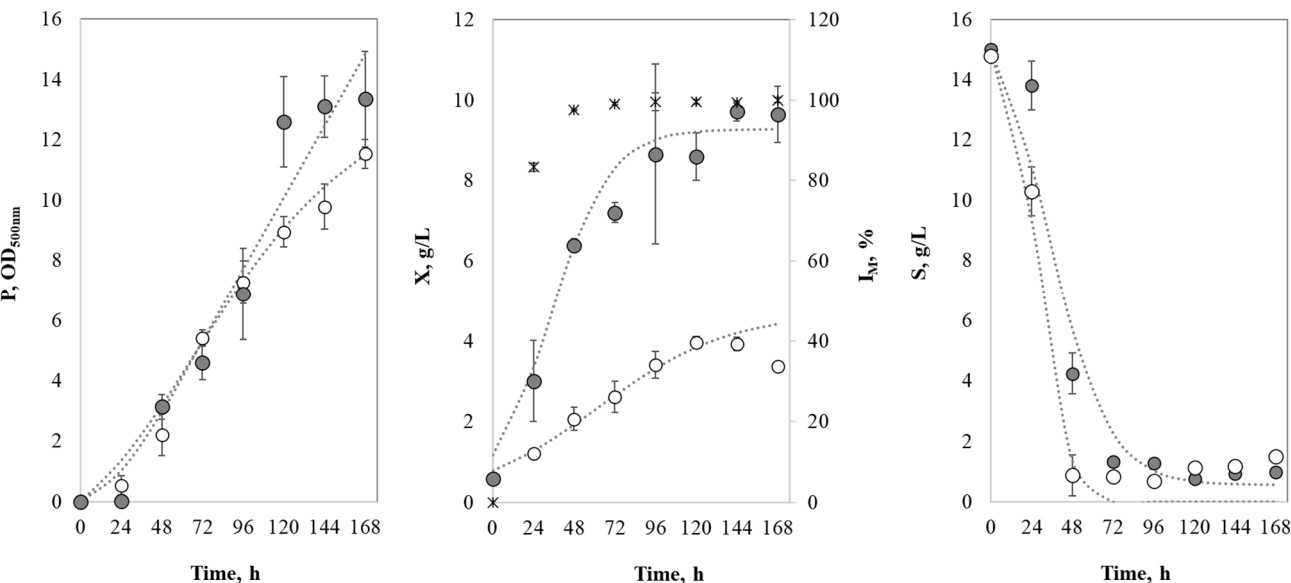

**Figure 5.** Kinetics of free (white circles) and immobilized cells (gray circles) of *Talaromyces atroroseus* GH2 in a single batch. Asterisks represent the immobilization percentage and dashed lines are the growth kinetic model's lines. Pigments, (P), biomass (X), immobilization capacity ($I_M$) and substrate (S). Error bars depict the standard deviation between three replicates.

**Table 4.** Growth, production and substrate consumption model parameters and goodness of fit.

| Parameter | Control | Immobilized |
|---|---|---|
| $x_o$, g/L | 0.79 | 0.81 |
| $x_{max}$, g/L | 3.75 | 9.32 |
| $S_o$, g/L | 15.00 | 15.00 |
| $P_o$, $OD_{500nm}$ | 0.00 | 0.00 |
| $\mu_o$, $h^{-1}$ | 0.03 | 0.06 |
| $\alpha$ | 2.53 | 0.22 |
| $\beta$ | 0.00 | 0.01 |
| $Y_{X/S}$, $g_{Biomass}/g_{Substrate}$ | 0.26 | 0.65 |
| $Y_{P/S}$, $OD/g_{Substrate}$ | 0.89 | 0.93 |
| $k_e$ | 0.09 | 0.00 |
| $R^2_X$ | 0.75 | 0.91 |
| $R^2_S$ | 0.95 | 0.96 |
| $R^2_P$ | 0.99 | 0.93 |

Instantaneous biomass concentration (x), production related to growth ($\alpha$), production related to maintenance ($\beta$), initial specific growth rate ($\mu_o$), yield of biomass per substrate ($Y_{x/s}$), yield of pigment per substrate ($Y_{p/s}$), cell maintenance coefficient ($k_e$), correlation coefficient for biomass production ($R^2_x$), correlation coefficient for substrate consumption ($R^2_s$), correlation coefficient for pigment production ($R^2_P$).

It can be observed that the microorganism showed a short lag phase (less than 24 h), and the stationary phase was reached at nearly 120 h in both cases. However, free cells presented a death cell phase, unlike the immobilized cells, which presented an extended stationary phase.

Regarding pigment production, immobilized *T. atroroseus* GH2 reached its peak of pigment production (13.01 ± 1.37 $OD_{500nm}$) at 120 h, while the free cells reached their peak of pigment production (10.09 ± 0.57 $OD_{500nm}$) at 168 h. This represents a reduction in the time of the whole process of 28% and an increment in the pigment production of 22% when using immobilized cells of *T. atroroseus* GH2 for pigment production. Biomass shows the total biomass production of both free and immobilized cells versus the progression of the immobilization capacity.

We previously suggested that the correct amount of immobilized cells can improve the production of pigments; concerning this, comparing the pigment production in both

free and immobilized cells, along with the biomass production, we can see a correlation between the time when the highest pigment production was reached and the time when biomass production was near its peak. A high concentration of cells in an immobilized system permits cells to improve the conversion of the carbon source into the metabolites of interest such as pigments [39].

As seen in Figure 5, when immobilized, *T. atroroseus* GH2 exhibits a noticeable major biomass production, reaching $9.32 \pm 0.51$ g/L, much more than the free cells that only reached $3.75 \pm 0.12$ g/L.

The improved conditions for the microorganism growth when using immobilized cells can also be identified by the initial specific growth rate ($\mu_o$). Immobilized cells showed twice the rate of free cells. Interestingly, in both cases, pigment production was associated with growth rather than cell maintenance (despite the difference in $\alpha$ and $\beta$ values). It has been extensively reported that fungal pigments are secondary metabolites produced under stress conditions and mostly produced during the stationary phase or at the end of the exponential growth [40–42]. The pigment production associated with growth in both cases could be attributed to the conditions utilized for inoculum preparation. The inoculum was prepared using PDA (with glucose added); thus, transferring the microorganism to a medium with a different carbon source than glucose (xylose) possibly stressed the microorganism.

Concerning substrate, as described in the modeling section, the substrate can be utilized for growth, metabolite production and cell maintenance. Biomass growth per substrate ($Y_{X/S}$) with immobilized cells was 2.5 times higher than that with free cells. As for pigment production per substrate ($Y_{P/S}$), obtained values were slightly similar. For free cells, the substrate was scarcely utilized for cell maintenance.

On the other hand, for immobilized cells, the cell maintenance coefficient ($k_e$) was zero. Such results are attributed to the growth phases presented in each case. While free cells showed the stationary and cell death phases, the immobilized cells seemed to be just reaching the stationary phase. In addition, it is visible how the free cells consumed the substrate faster than the immobilized cells. Free cells consumed practically all the substrate at 48 h; the immobilized cells, at around 96–120 h. The lower substrate consumption rate by immobilized cells could be due to a lower oxygen availability within the support.

Additionally, it was observed that despite the high correlation coefficient for substrate consumption ($R^2 > 0.95$), the model lines (dashed lines in Figure 5) failed to describe the experimental data accurately.

We attributed those results to the presence of ethanol. Previous studies from the research group demonstrated that adding ethanol to the culture medium promoted the production of pigments by *Talaromyces atroroseus* GH2 [43]. Since the late 1990s, it has been explained how filamentous fungi, as regards pigment production, have the ability to uptake ethanol either produced by the microorganism itself during the fermentation or already present in the fermentation media as a secondary carbon source, provoking a shift in the metabolism from growing to maintenance and thus favoring the production of pigments [40].

Most recently, it was demonstrated that *Talaromyces atroroseus* GH2 can consume ethanol [43]. Thus, this can explain why pigment production increases exponentially once the main carbon source, xylose, is fully consumed. Double substrate kinetics models might provide a better understanding of how nutrient conditions affect the performance of immobilized cells of *Talaromyces atroroseus* GH2 in producing pigments and the future development of a continuous or a semi-continuous process.

As regards biomass immobilization, results showed that practically 100% of biomass immobilization occurred in the early stages of the process, 48 h, and remained stable throughout the process. In summary, aiming at pigment production, 120 h seemed to be the ideal time for the immobilized cells of *Talaromyces atroroseus* GH2 because, at that time, the peak of pigment production was reached, total biomass production was practically at its maximum, the whole biomass remained immobilized and the main substrate was depleted. Hence, 120 h was selected for the cell recycling assessment studies.

### 3.3.2. Cell Recycling Assessment during Consecutive Batches

Based on previous section results, 120 h per batch was selected for immobilized biomass re-utilization in consecutive batches and the evaluation of the process kinetics. The immobilized biomass was recycled three times. Figure 6 shows the kinetics of pigment production, substrate consumption and biomass growth results.

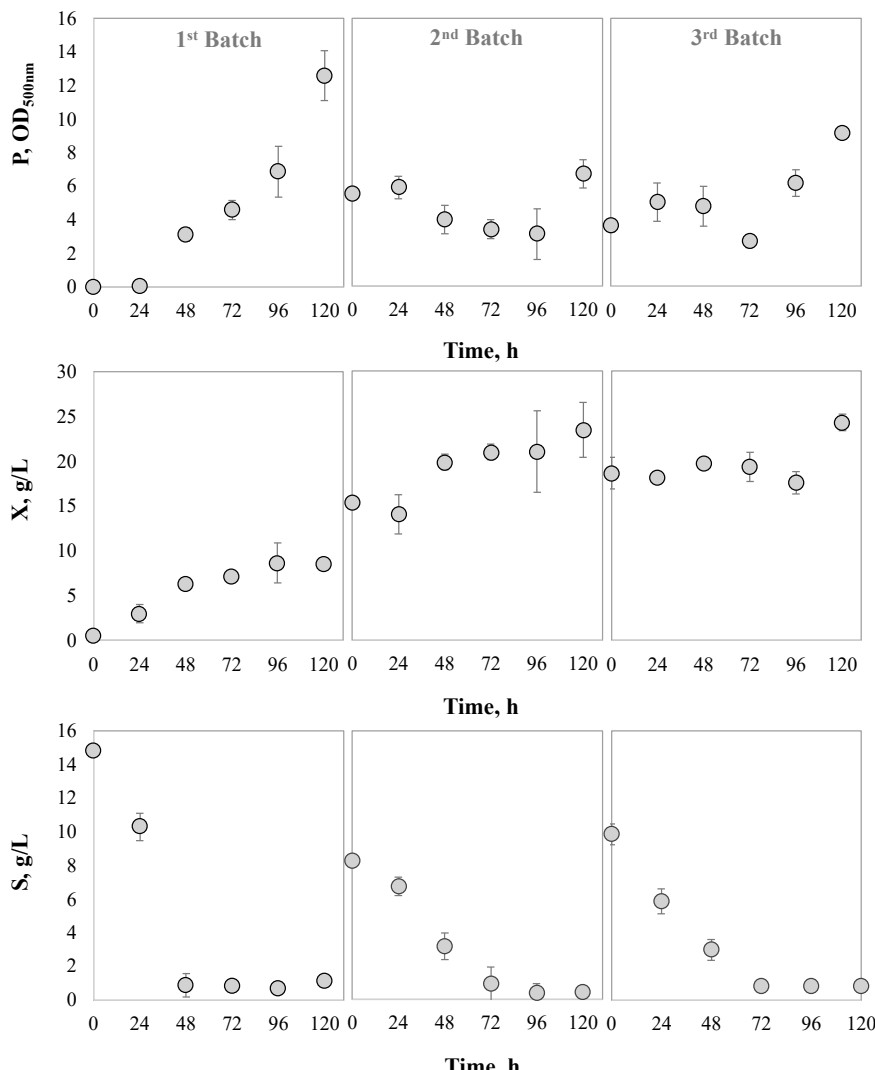

**Figure 6.** Kinetics of the immobilized cells of *Talaromyces atroroseus* GH2 in three consecutive batches. Pigments (P), biomass (X), substrate (S). Error bars depict the standard deviation between three replicates.

In the first batch, pigment production followed the same tendency seen in the previous sections, reaching the production peak at 120 h. Once the biomass was first recycled, the pigment production behavior changed in response to the environmental changes. As seen in Figure 6, at 0 h, the pigment measurement started at $5.54 \pm 0.32$ $OD_{500nm}$ but then decreased and kept that tendency until 120 h, when the pigment production began, thus following the same trend described before. Those results were attributed to pigment desorption and re-absorption phenomena exhibited by both the immobilization support and the biomass. In the third batch, the pigment production tendency seems similar but more unstable.

Biomass growth followed a marked trend; it kept growing throughout the process. Finally, the immobilization capacity at the end of each batch remained over 98%, as follows: batch 1, $99.45 \pm 0.23$; batch 2, $98.25 \pm 1.43$; and batch 3, $98.43 \pm 0.71$.

Regarding substrate consumption, the tendency for the first batch was the same as before; the main substrate is consumed right after inoculation, and it is fully depleted at 72 h. The second and third batches had different behavior. The substrate consumption was slower in comparison with the first batch; the substrate was not depleted until 96 h for the second batch and 72 h for the third batch even though a lower xylose concentration was used. Recently, it was shown for the production of pigments by immobilized mycelium of *Penicillium brevicompactum* that, due to the formation of a dense coating, the substrate was not consumed [41]. In this study, the microorganism almost depleted the substrate, but it was shown that *T. atroroseus* GH2 behaved differently regarding pigment production and substrate consumption in consecutive batches. It was clear how the main substrate must be fully depleted before the pigment production begins.

This type of secondary metabolite is secreted in the media when the environmental conditions are harsh [42], so exposing *T. atroroseus* GH2 to fresh media, rich in nutrients, can halt the production of pigments and provoke a re-intake of the pigments already produced. It is possible that the microorganism was well adapted to the medium containing xylose; thus, exposing *T. atroroseus* GH2 to a fresh medium for the new batch can also explain the tendency of the biomass to keep growing, reaching levels of biomass that the immobilization support cannot hold.

## 4. Conclusions

At first, the immobilization of *Talaromyces atroroseus* GH2 did not represent a significant increment in pigment production compared to free cells in a single batch. However, it was demonstrated how it was possible to reuse the immobilized cells of *T. atroroseus* GH2 to develop a continuous process with significant advantages.

The application of the Taguchi method led us to identify the significance of the effects of the evaluated factors on *T. atroroseus* GH2 growth and the production of pigments. At improved immobilization conditions, pigment production was higher than that observed using free cells, and this high pigment production was accompanied by a long-lasting immobilization activity. Process kinetics showed that the production could continue for three batches and was limited by excessive microorganism growth.

The immobilization technique is advantageous for easing the liquid–solid separation from the reaction mixture, thus simplifying the downstream process during pigment production. A continuous process for pigment production has not been developed; hence, this research field is a promising niche. Nonetheless, more studies about immobilization and nutrient requirements in consecutive batches are still needed. The promising results pave the way for further investigation to improve the responses and develop a robust continuous process for pigment production by immobilized cells of *Talaromyces atroroseus* GH2.

**Author Contributions:** Experiment performance and initial draft writing, J.P.R.-S.; experiment design, data analysis and research supervision, L.M.-O.; manuscript review and editing, D.G.; manuscript writing and editing, L.D.; conceptualization and funding acquisition, J.C.M. All authors have read and agreed to the published version of the manuscript.

**Funding:** This research was funded by Consejo Nacional de Ciencia y Tecnología, grant number [887396].

**Institutional Review Board Statement:** Not applicable.

**Informed Consent Statement:** Not applicable.

**Data Availability Statement:** The data will be available from the corresponding author upon a reasonable request.

**Acknowledgments:** Author J.P.R.-S. expresses their gratitude to the Mexican National Council for Science and Technology (CONACYT-México) for the grant received to pursue his postgraduate studies.

**Conflicts of Interest:** The authors declare no conflict of interest.

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
