# Peer review of "Production of Pigments under Submerged Culture through Repeated Batch Fermentation of Immobilized Talaromyces atroroseus GH2"

_fermentation, doi:10.3390/fermentation9020171_

Round 1
Reviewer 1 Report
The authors may know that the natural pigments produced by Talaromyces spp. are almost identical to the ingredients of Monascus pigments (MPs) that have been used as food colorants in China for about 2000 years, and the biosynthetic gene clusters and biosynthetic pathways of both pigments are almost identical. In addition, MPs have always been legal food colorants in China. Whether the pigments from Talaromyces spp.can be used as food colorants in Europe at present, and what is the future situation? I think the authors should have some description about these in the manuscript. Moreover, as far as I know, most of the components of MPs exist in the Monascus cells. At present, the industrial production of MPs in China is mainly to extract pigment products from the cells. But this manuscript describes that the pigment from Talaromyces spp. is mainly secreted to the outside of the cells. Please provide relevant data or references.
Reviewer 2 Report
Manuscript “Production of pigments under submerged culture through 2 repeated batch fermentation of immobilized Talaromyces atro-3 roseus GH2”
Abstract:
It is important to make clear that the kinetics studies were performed for the production process and not to biomass immobilization process.
I suggest to change the sentence “optimized conditions” since the Taguchi method is more exploratory than statistical tool for improve a process.
How the process of immobilization can help in the downstream processing?
Key-words: I suggest to change “continuous process”.
Introduction
I suggest to improve the explanation about the necessity to replace synthetic colorants. Not all synthetic colorants are danger and the sentence improve is too generic.
The authors employ to terms: colorants and pigments. Are they synonyms?
Since the pigments produced by Talaromyces atroseus are extracellular. How the immobilization process can reduce the time for downstream processing? Moreover, to recover the cells a specific process has to be employed?
Materials and methods
Insert the meaning of PDB.
Line 109 is PDA or PDB?
Why the density of support was different for each support?
It is confusing how was performed the support selection for immobilization. Was determined the number of spores of cell viability after the immobilization? How was made the immobilized biomass transferring?
How was calculated the immobilized biomass and immobilization capacity? Insert the equation.
Standardize using hours or days.
Line 165, the authors say that only extracellular colorants were considered but how were the cells after the bioprocess? Previously, the authors wrote that the pigments were extracellular and at this line made this consideration. So, there are pigments intracellular and extracellular?
Since absorbance is an indirect method, I suggest to not use the term “concentration” instead use “amount of pigment”.
If the cells were used in different batch, how was calculated the pigment yield?
Results and discussion
It is interesting to make an introduction in the paragraph before present the results.
Insert the result of pigment production with free cells in Table 1. Moreover, insert the meaning of abbreviations as footnote in the Table 1.
Was the pigment production with immobilized cells performed in triplicate or the measurements? Please insert the statistical analysis in the results. The error bar of pigment production mainly with he supports A and B are high.
It is not clear how was the immobilization process. I suggest to make a Transmission electronic microscopy analysis to explain how the immobilization process occurred. Was the support selected based in the pigment production?
Was the support volume calculated based on size of Erlenmeyer flasks or culture media volume?
Since the process of transferring oxygen from the medium to the interior of the cell is a fundamentally important parameter to produce metabolites by microorganisms, how did the immobilization process affect this process?
Is the growth of the microorganism limited by immobilization?
How was it verified that the immobilized cells could not be used? Cell viability or was there a blocking of fixed assets?
Line 453-455, the authors say that the immobilized cells can improve the conversion of carbon source in metabolites. In a biochemical point of view, why this happens? Are there some interactions between the cells and the support?
In conditions of stress, generally, the cells do not growth, however, the cells kept growing in the presence of xylose. Please, improve the explanation considering the cell biochemistry.
Insert the meaning of error bars in the Figures.
Reviewer 3 Report
This manuscript by Ruiz-Sánchez et al. studied the production of pigments by fermentation of immobilized Talaromyces atroroseus GH2. They evaluated three support materials for pigment production and immobilization capacity. And optimized immobilization conditions by determining the effect of four factors (i.e., inoculum concentration, support density, working volume, and support volume) on fungal immobilization and pigment production. They also evaluated the process kinetics in either a single batch manner or a repeated batch.
Please address the following comments prior to publication:
1. Line 101 and 102: Please define PDA and PDB.
2. Line 104: Please change MgSO4*7H2O and FeSO4*7H2O to MgSO4·7H2O and FeSO4·7H2O, respectively.
3. Line 195: Please define uo and Xmax in equation 1.
4. 3.2 Immobilization conditions optimization section: Please describe the support used in this section.
5. Line 426-428 and Figure 4: Please show R2 in figure 4.
